# LEARNING WHAT AND WHERE TO ATTEND

**Drew Linsley, Dan Shiebler, Sven Eberhardt and Thomas Serre**
Department of Cognitive Linguistic & Psychological Sciences
Carney Institute for Brain Science
Brown University
Providence, RI 02912
`{drew_linsley,thomas_serre}@brown.edu`

## ABSTRACT

Most recent gains in visual recognition have originated from the inclusion of attention mechanisms in deep convolutional networks (DCNs). Because these networks are optimized for object recognition, they learn where to attend using only a weak form of supervision derived from image class labels. Here, we demonstrate the benefit of using stronger supervisory signals by teaching DCNs to attend to image regions that humans deem important for object recognition. We first describe a large-scale online experiment (ClickMe) used to supplement ImageNet with nearly half a million human-derived "top-down" attention maps. Using human psychophysics, we confirm that the identified top-down features from ClickMe are more diagnostic than "bottom-up" saliency features for rapid image categorization. As a proof of concept, we extend a state-of-the-art attention network and demonstrate that adding ClickMe supervision significantly improves its accuracy and yields visual features that are more interpretable and more similar to those used by human observers.

## 1 INTRODUCTION

Attention has become the subject of intensive research within the deep learning community. While biology is sometimes mentioned as a source of inspiration (Stollenga et al., 2014; Mnih et al., 2014; Cao et al., 2015; You et al., 2016; Chen et al., 2017; Wang et al., 2017; Biparva and Tsotsos, 2017), the attentional mechanisms that have been considered remain limited in comparison to the rich and diverse array of processes used by the human visual system (see Itti et al., 2005, for a review). In addition, whereas human attention is controlled by varying task demands, attention networks used in computer vision are solely optimized for object recognition. This means that, unlike infants who can rely on a myriad of visual cues and supervision to learn to focus their attention (Itti et al., 2005), DCNs must solve this challenging problem with weak supervisory signals derived from statistical associations between image pixels and class labels. Here, we investigate how explicit human supervision – teaching DCNs what and where to attend – affects their performance and interpretability.

### 1.1 RELATED WORK

**Attention models in human vision**   Attention has been extensively studied from a computational neuroscience perspective (see Itti et al., 2005, for a review). One key motivation for the present work includes the presence of complementary pathways for visual attention, which operate on global versus local features. *Global features* encode scene layout or "gist", and are hypothesized to capture the contextual information which guides visual processing to task-relevant image locations (Oliva and Torralba, 2007). This is most similar to the feature-based attention that is used in state-of-the-art networks (Bell et al., 2016; Wang et al., 2017; Hu et al., 2017). *Local features* in attention models are assumed to drive a separate form of attention known as visual saliency (Itti and Koch, 2001). A widely held belief is that visual saliency provides a task-independent topographical encoding of feature conspicuity in a visual scene. It has been proposed that the rapid convergence of these two pathways acts as an efficient shortcut for filtering clutter from object detection processes (Torralba et al., 2006).

**Attention networks in computer vision** An extensive body of work aims at explicitly predicting human eye fixations and/or detecting objects (see Nguyen et al., 2018, for a review). In addition, recent research on image categorization has focused on integrating attention modules within end-to-end trainable DCNs. These attention modules fall into two broad categories that are conceptually similar to the global and local pathways studied in visual neuroscience. *Feature-based attention* (also called "channel-wise" attention) involves learning a task-specific feature modulation that is applied across an entire scene to refine feature maps. *Spatial attention* is a complementary form of attention which involves learning a spatial mask that enhances/suppresses the activity of computational units inside/outside an attention "spotlight" positioned in a scene. This is done according to the spatial location of individual network units independently of their feature tuning. Such mechanisms have been shown to significantly improve performance on visual question answering and captioning (e.g., Nam et al., 2017; Patro and Namboodiri, 2018). Here, we combine spatial- and feature-based attention into a single mask that modulates feature representations, similarly to state-of-the-art approaches (e.g., Chen et al., 2017; Wang et al., 2017; Biparva and Tsotsos, 2017; Park et al., 2018; Jetley et al., 2018), with a formulation that additionally supports supervision from human annotations.

**Humans-in-the-loop computer vision** A central goal of the present study is to leverage human supervision to co-train an attention network. Previous work has shown that it is possible to augment vision systems with human perceptual judgments on many difficult recognition problems (e.g., Vondrick et al., 2015; Shanmuga Vadivel et al., 2015; Kovashka et al., 2016; Deng et al., 2016). In particular, online games constitute an efficient way to collect high-quality human ground-truth data (e.g., von Ahn et al., 2006; Deng et al., 2016; Das et al., 2016; Linsley et al., 2017). In a two-player game, an image may be gradually revealed to a "student" tasked to recognize it, by a remote "teacher" who draws "bubbles" on screen to unveil specific image parts (Linsley et al., 2017). The game-play mechanics introduced by assembling teams of players reduces annotation noise in these games, but also limits their suitability for large-scale data collection. This limitation is aleviated in one-player games (Deng et al., 2013; 2016; Das et al., 2016), where a player may be asked to sharpen regions of a blurred image to answer questions about it. The game introduced here differs from these earlier studies (Linsley et al., 2017) in that it has a human player collaborate with a DCN to discover important visual features for recognition at ImageNet scale.

**Attention datasets** Recording eye fixations while viewing a stimulus is a classic way to explore visual attention (e.g., Judd et al., 2009). It is, however, difficult and costly to acquire large-scale eye tracking data, leading researchers to instead track computer mouse movements during task-free image viewing. Maps collected from individual observers can then be combined into a single "bottom-up" saliency map, as is done with the popular Salicon dataset (Jiang et al., 2015). Still, Salicon contains saliency maps for 10k images, which is relatively small for training deep networks on object recognition. We instead describe a gamification procedure used to collect nearly a half-million "top-down" (task-driven) attention maps over several months. As we demonstrate through psychophysics, our top-down attention maps also better reflect human recognition strategies than Salicon (section 2.2), as they are collected while human observers search for visual features that are informative for object categorization.

## 1.2 CONTRIBUTIONS

Our contributions are three-fold: (i) We describe a large-scale online experiment `ClickMe.ai` used to supplement ImageNet with nearly a half-million top-down attention maps derived from human participants. These maps are validated using human psychophysics and found to be more diagnostic than bottom-up attention maps for rapid visual categorization. (ii) As a proof of concept, we describe an extension of the leading squeeze-and-excitation (SE) module: the global-and-local attention (GALA) module, which combines global contextual guidance with local saliency and substantially improves accuracy on ILSVRC12. (iii) We find that incorporating ClickMe supervision into GALA training leads to an even larger gain in accuracy while also encouraging visual representations that are more interpretable and more similar to those derived from human observers. By supplementing ImageNet with the public release of ClickMe attention maps, we hope to spur interest in the development of network architectures that are not only more robust and accurate, but also more interpretable and consistent with human vision.

## 2 CLICKME.AI

A large-scale effort is needed to gather sufficient attention annotations for training neural network models. Our starting point for this endeavor is the Clicktionary game that we introduced in (Linsley et al., 2017), which pairs online human participants to cooperatively annotate object images. This two-player game was successfully used to collect attention maps for a few hundred images, but we found it impossible to scale beyond that because of the challenge of matching pairs of players. These limitations prompted us to develop `ClickMe.ai`, an adaptation of Clicktionary into a single-player game that supports large-scale data acquisition.

ClickMe consists of rounds of game play where human participants play with DCN partners to recognize images from the ILSVRC12 challenge. Players viewed one object image at a time and were instructed to use the mouse cursor to "paint" image parts that were most informative for recognizing its category (written above the image). Once the participant clicked on the image, $14 \times 14$ pixel bubbles were placed wherever the cursor went until the round ended (Fig. 1). Having players annotate in this way forced them to carefully monitor their bubbling while also preventing fastidious strategies that would produce sparse salt-and-pepper types of maps.

As players bubbled image regions they deemed important for recognition, a DCN tried to recognize a version of the image in which only these bubbled parts were visible. We tried to make the game as entertaining and fast-paced as possible to maximize the number of rounds played by the human players. Hence, we nearly doubled the size of the bubbled regions shown to the DCN ($21 \times 21$ pixels) to make sure that the objects were recognizable to the DCN within a few seconds of play (Fig. 1). The reasons for keeping a DCN in the loop (as opposed to a random timer) were (1) to make the game entertaining, and (2) to discourage human players from bubbling random locations in an image which are potentially unrelated to the object. Indeed, we have incorporated all the images used in Clicktionary and found that ClickMe and Clicktionary maps are strongly correlated (see Appendix; (Linsley et al., 2017)). This suggests that replacing one of the human players with a DCN did not bias the collected annotations, but it did allow us to collect human data at a scale suitable for co-training a DCN with human attention annotations.[1]

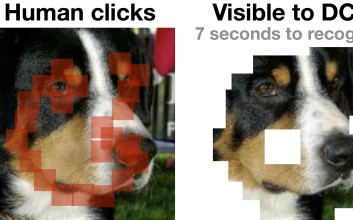

**Human clicks**   **Visible to DCN**
7 seconds to recognize

Figure 1: The ClickMe interface for human participants and their DCN partners. Participants select important image parts with their mouse by "painting" translucent bubbles on screen. At the same time, image parts of roughly double the size are shown to a DCN partner, tasked to recognize the image. Each round lasts until the DCN recognizes the object or if 7 seconds have passed (the latter occurred 47% of the time).

A timer controlled the number of points participants received per image, and high-scoring participants were awarded prizes (see Appendix). Points were calculated as the proportion of time left on the timer after the DCN achieved top-5 correct recognition for the image. If the player could not help the DCN recognize the object within 7 seconds, the round ended and no points were awarded.

### 2.1 GAME STATISTICS

Data collection efforts on `ClickMe.ai` drew 1,235 participants (unique user IDs) who played an average of 380 images each. In total, we recorded over 35M bubbles, producing 472,946 ClickMe maps on 196,499 unique images randomly selected from our preparation of ILSVRC12 (see Appendix for details). All ClickMe maps were used in subsequent experiments, regardless of the ability of the DCN partner to correctly identify them within the time limit of a round. Fig. 2A shows sample ClickMe maps where pixel opacity reflects the likelihood that it was bubbled by participants. ClickMe maps often highlight local image features and emphasize specific object parts over others. For instance, ClickMe maps for animal categories (Fig. 2A, top row) are nearly always oriented towards facial components even when these are not prominent (e.g., snakes). In general, we also found that ClickMe maps for inanimate objects (Fig. 2A, bottom row) exhibited a front-oriented bias, with distinguishing parts such as engines, cockpits, and wheels receiving special attention. Additional game information, statistics and ClickMe maps are available as Appendix.

---

[1]See Appendix for an extended discussion on why human observers likely adopted strategies for selecting object features relevant to their recognition instead of performing a sensitivity analysis on their DCN partners.

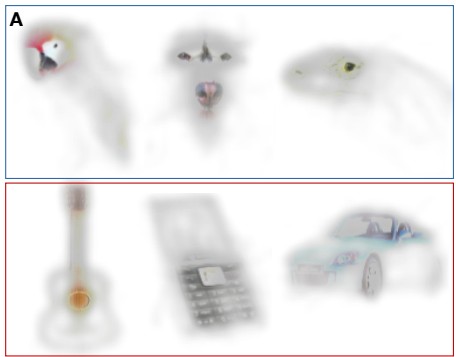 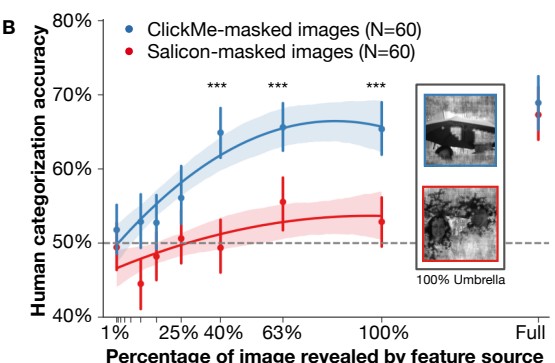

Figure 2: **(A)** A representative selection of ILSVRC12 images and their ClickMe maps. The transparency channel of these images is set to the proportion of ClickMe bubbles for that location across participants, making feature visibility reflect importance. Animals are outlined in blue and non-animals in red. **(B)** Features identified in "top-down" ClickMe maps are more diagnostic for object recognition than those identified in "bottom-up" Salicon maps. A rapid visual categorization experiment compared human performance in detecting animals when features were revealed according to ClickMe maps (blue curve) or Salicon maps (red curve). ClickMe-masked and Salicon-masked image exemplars are shown for the condition in which 100% of important features are visible, demonstrating how "bottom-up" saliency is not necessarily relevant to the task. For clarity, we omitted data between 1-10% of features visible from this plot where accuracy was chance for participants of both groups. Error bars are S.E.M. ***: $p < 0.001$ (statistical testing with randomization tests detailed in the Appendix).

## 2.2 CLICKME AND OBJECT RECOGNITION

Rapid vision experiments have classically been used in visual neuroscience to probe visual responses while controlling for feedback processes (Thorpe et al., 1996; Serre et al., 2007; Eberhardt et al., 2016). Here, we devised such a rapid vision categorization experiment to compare the contribution of top-down ClickMe features for object recognition with features derived from bottom-up image saliency (Fig. 2B), closely following the approach of (Eberhardt et al., 2016). If human participants were more effective at recognizing object images based on ClickMe features than bottom-up saliency features, we reasoned it would validate the ClickMe approach and demonstrate the relevance of the collected feature importance maps to object recognition.

Our design of this experiment followed the rapid visual categorization paradigm used in (Eberhardt et al., 2016) where stimuli are quickly flashed and responses are forced within 550 ms of stimulus onset (see Appendix for details). We recruited 120 participants from Amazon Mechanical Turk (www.mturk.com), each of whom viewed images that were masked to reveal a randomly selected amount of their most important visual features. Participants were organized into two groups ($N = 60$ participants in each): one viewed images that were masked according to ClickMe maps, and the other viewed images that were masked according to bottom-up saliency.

We tested participant responses on 40 target (animal) and 40 distractor (non-animal) images gathered from the Salicon (Jiang et al., 2015) subset of the Microsoft COCO 2014 (Lin et al., 2014) because it includes bottom-up saliency maps derived from human observers. Images were presented to human participants either intact or with a phase-scrambled perceptual mask which selectively exposed their most important visual features according to attention maps derived from either ClickMe or Salicon. Attention maps from both resources were preprocessed so that there was spatial continuity for the pixels covering their most-to-least important visual features. This ensured that a low-level difference between attention maps, such as the presence of many discontinuous visually important regions, did not bias participants' responses in this rapid categorization experiment. Preprocessing was done with a novel "stochastic" flood-fill algorithm that relabeled attention map pixels with a score that combined their distance from the most important pixel with their labeled importance (see Appendix for details). Next, we created versions of each image that revealed between 1% and 100% (at log-scale spaced intervals) of its most important pixels.

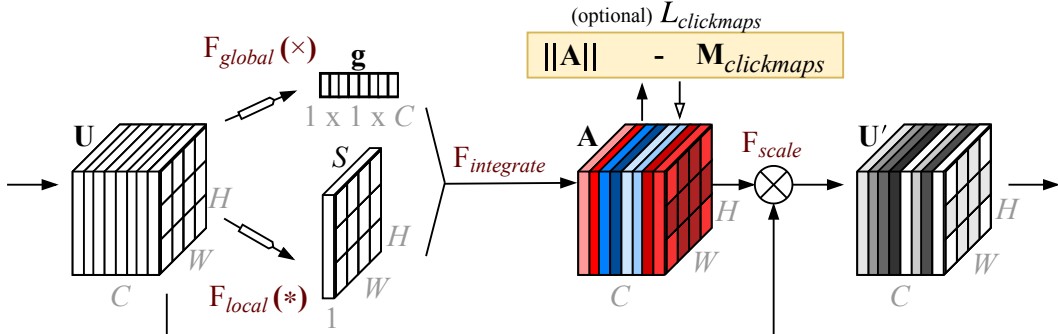

Figure 3: The global-and-local (GALA) module learns to combine local saliency with global contextual signals to guide attention towards image regions that are diagnostic for object recognition. The diagram here depicts a GALA module applied to the feature activity from a convolutional layer in a deep network model, **U**. Information in the diagram flows from left to right, and **U** is processed with separate local ($F_{local}$) and global ($F_{global}$) operators to derive the local- and global-attention masks, which is integrated into the attention activity **A**. Attention is applied to the original activity, **U**, with elementwise multiplication ($F_{scale}$), to yield **U**′. A GALA module can optionally be supervised by ClickMe maps ($\mathbf{M}_{clickmaps}$ in the yellow box) with an additional loss $\mathbf{L}_{clickmaps}$ that evaluates the distance between these maps and **A**. Minimizing this loss encourages GALA to select visual features favored by humans (see Section 4 for details).

This experiment measured how increasing the proportion of important visual features from either ClickMe or Salicon attention maps influenced behavior (see insets in 2B for examples of images where 100% of ClickMe or Salicon features were revealed). Experimental results are shown in Fig. 2B: Human observers reached ceiling performance when 40% of the ClickMe features were visible (6% of all image pixels). In contrast, human observers viewing images masked according to Salicon required as much as 63% of these features to be visible, and did not reach ceiling performance until the full image was visible (accuracy measured from different participant groups). These findings validate that visual features measured by ClickMe are distinct from bottom-up saliency and sufficient for human object recognition.

## 3 PROPOSED NETWORK ARCHITECTURE

We designed the *global-and-local attention* (GALA) block as a circuit for learning complex combinations of local saliency and global contextual modulations that can optionally be supervised by ClickMe maps. The rational for this design, in particular the parallel attention pathways, is grounded in visual neuroscience and detailed in section 1.1. Below we sketch the main computational elements of the architecture, and describe the process by which it is used to modulate activity at a layer in a DCN model (see Fig. 3 for a high-level overview of GALA attention transforming the DCN layer activity **U** into **U**′ with the derived attention activity **A**).

### 3.1 ARCHITECTURE

GALA modulates an input layer activity **U** with an attention mask **A** of the same dimension as the input, which captures a combination of global and local forms of attention. Here, the spatial height, width, and number of feature channels are denoted as $H$, $W$, and $C$ s.t. $\mathbf{U}, \mathbf{A} \in \mathbb{R}^{H \times W \times C}$.

We begin by describing global attention in the GALA, which is based on the SE module (Hu et al., 2017). Global attention is denoted $F_{global}$ in our model and yields the global feature attention vector $\mathbf{g} \in \mathbb{R}^{1 \times 1 \times C}$ (Fig. 3). This procedure involves two steps: first, per-channel "summary statistics" are calculated; and second, a multilayer perceptron (MLP) applies a non-linear transformation to these statistics. Summary statistics are computed with a global average applied to the feature maps $\mathbf{U} = [\mathbf{u}_k]_{k=1...C}$ to yield the vector $\mathbf{p} = (p_k)_{k=1...C}$. That is, $\mathbf{p_k} = \frac{1}{WH} \sum_{x=1}^{W} \sum_{y=1}^{H} \mathbf{u}_{kxy}$. This is

followed by a two-layer MLP to shrink and then expand the dimensionality of $\mathbf{p}$. An intervening nonlinearity enables the module to learn complex dependencies between channels. The shrinking operation of this MLP ("squeeze" from Hu et al., 2017) is applied to the vector $\mathbf{p}$ by the operator $W_{shrink} \in \mathbb{R}^{\frac{c}{r} \times C}$, mapping it into a lower dimensional space. This is followed by an expansion operation ("excitation" from Hu et al., 2017) $W_{expand} \in \mathbb{R}^{C \times \frac{c}{r}}$ (bias terms are omitted for simplicity) back to the original, higher dimensional space s.t. $\mathbf{g} = W_{expand}(\delta(W_{shrink}(\mathbf{p})))$. We set $\delta$ to be a rectified linear function (ReLU) and the dimensionality "reduction ratio" $r$ of the shrinking operation to 4 as in the original work.

In addition to the SE/global attention module, we consider a local saliency module $\mathrm{F}_{local}$ for computing the local feature attention map $S$ (Fig. 3) as $S = \mathbf{V}_{collapse} * (\delta(\mathbf{V}_{shrink} * \mathbf{U}))$. Here, convolution is denoted with $*$, $\mathbf{V}_{shrink} \in \mathbb{R}^{1 \times 1 \times C \times \frac{c}{r}}$, and $\mathbf{V}_{collapse} \in \mathbb{R}^{1 \times 1 \times \frac{c}{r} \times 1}$. This is reminiscent of the local computations performed in computational-neuroscience models of visual saliency to yield per-channel conspicuity maps that are then combined into a single saliency map (Itti and Koch, 2001).

Outputs from the local and global pathways are integrated with $\mathrm{F}_{integrate}$ to produce the attention volume $\mathbf{A} \in \mathbb{R}^{H \times W \times C}$. Because it is often unclear how tasks benefit from one form of attention versus another, or whether task performance would benefit from additive vs. multiplicative combinations, $\mathrm{F}_{integrate}$ learns parameters that govern these interactions. The vector $(a_c)_{c \in 1..C}$ controls the additive combination of $S$ and $\mathbf{g}$ per-channel, while $(m_c)_{c \in 1..C}$ does the same for their multiplicative combination. In order to combine attention activities $\mathbf{g}$ and $S$, they are first tiled to produce $\boldsymbol{G}^*, \boldsymbol{S}^* \in \mathbb{R}^{H \times W \times C}$. Finally, we calculate the attention activities of a GALA module as $\mathbf{A}_{h,w,c} = \zeta \left( a_c(\boldsymbol{G}^*_{h,w,c} + \boldsymbol{S}^*_{h,w,c}) + m_c(\boldsymbol{G}^*_{h,w,c} \cdot \boldsymbol{S}^*_{h,w,c}) \right)$, where the activation function $\zeta$ is set to the $\tanh$ function, which squashes activities in the range $[-1, 1]$. In contrast to other bottom-up attention modules, which use a sigmoidal function to implement a soft form of excitation and inhibition, our selection of $\tanh$ gives a GALA module the additional ability to dis-inhibit bottom-up feature activations from $\mathbf{U}$ and flip the signs of its individual unit responses. Attention is applied by $\mathrm{F}_{scale}$ as $\mathbf{U}' = \mathbf{U} \odot \mathbf{A}$.

## 3.2 RESNET-50 IMPLEMENTATION

We validated the GALA approach by embedding it within a ResNet-50 (He et al., 2016). We identified 6 mid- to high-level feature layers in ResNet-50 to use with GALA (layers 24, 27, 30, 33, 36, 39; each of which belong to the same ResNet-50 processing block), since these will in principle encode visual features that are qualitatively similar to the object-parts highlighted in ClickMe maps (see Fig. 2A for an example of such parts). Each GALA module was applied to the final activity in a dense path of a residual module in ResNet-50. At this depth in the ResNet, GALA attention activity maps had a height and width of $14 \times 14$. The residual layer's "shortcut" activity maps were added to this GALA-modulated activity to allow the ResNet to flexibly adjust the amount of attention applied to feature activities. The accuracy of our re-implementations of ResNet-50 (He et al., 2016) and SE-ResNet-50 (Hu et al., 2017) trained on ILSVRC12 are on par with published results (Appendix Table 2). Incorporating our proposed GALA module into the ResNet-50 (*GALA-ResNet-50 no ClickMe*) offers a small improvement over the SE-ResNet-50. In section 4 we will demonstrate that the benefits of GALA are more evident on smaller datasets and when we add attention supervision.

## 4 CO-TRAINING GALA WITH CLICKME MAPS

To this point, we have described the ClickMe dataset, which contains human-derived feature importance maps for ILSVRC12. We have also introduced the GALA module for learning local- and global-attention during object recognition. In this section we describe a method for supervising GALA with ClickMe maps and its effect on model performance and interpretability.

We use ClickMe maps to supervise GALA modules by introducing an additional loss. Let $\mathcal{L}_C$ denote the cross-entropy between activity from model $M$ with input $\mathbf{X}$ and class label $y$. We combine this with a loss which describes the distance between GALA module activity $\mathbf{A}^l(\mathbf{X})$ at layer $l \in \mathbf{L}$ and the ClickMe map for the image $\mathbf{X}$:

$$\mathcal{L}_T(\mathbf{X}, y) = \mathcal{L}_C(M(\mathbf{X}), y) + \lambda \sum_{l \in \mathbf{L}} \left\| \frac{R^l(\mathbf{X})}{\|R^l(\mathbf{X})\|_2} - \frac{\mathbf{A}^l(\mathbf{X})}{\|\mathbf{A}^l(\mathbf{X})\|_2} \right\|_2 \tag{1}$$

|                          | top-1 err | top-5 err | maps      |
|--------------------------|-----------|-----------|-----------|
| SE-ResNet-50             | 66.17     | 42.48     | 64.36**   |
| ResNet-50                | 63.68     | 40.65     | 43.61     |
| GALA-ResNet-50 no ClickMe| 53.90     | 31.04     | 64.21**   |
| GALA-ResNet-50 w/ ClickMe| **49.29** | **27.73** | **88.56**** |

Table 1: Top-1 and top-5 classification error of networks along with the fraction of human ClickMe map variability explained by their features (maps; 100 corresponds to the average similarity between human ClickMe maps). Performance is reported on the test set of ClickMe. ** denotes $p$ <0.01 (statistical testing captures the proportion of image feature maps that exceed the null inter-participant reliability score; see Appendix for details).

Minimizing the global loss $\mathcal{L}_T$ jointly optimizes a model for object classification and predicting ClickMe maps from input images (the latter loss is referred to in Fig. 3 as $L_{clickmaps}$). The latter loss is scaled by the hyperparameter $\lambda$, which is selected according to experiments discussed in section 4.2. To prepare ClickMe maps for this loss, they are resized with bicubic interpolation to be the same height and width as a GALA module activity $\mathbf{A}^l(\mathbf{X})$ at layer $l \in \mathbf{L}$, the set of layers where the GALA module is applied. We denote the resized ClickMe map for this input as $R^l(\mathbf{X})$. We also reduce the depth of GALA activity tensors $\mathbf{A}^l(\mathbf{X})$ to 1 by setting them to their column-wise $L_2$ norm. Finally, units in $R^l(\mathbf{X})$ (ClickMe maps) and $\mathbf{A}^l(\mathbf{X})$ (GALA activity) are transformed to the same range by normalizing each by their $L_2$ norms.

## 4.1 MODEL EVALUATION

We evaluated our approach for supervising GALA with ClickMe maps by partitioning the ClickMe dataset into separate folds for training, validation, and testing. We set aside approximately 5% of the dataset for validation (17,841 images and importance maps), another 5% for testing (17,581 images and importance maps), and the rest for training (329,036 images and importance maps). Each split contained exemplars from all 1,000 ILSVRC12 categories. Training and validation splits were used in the analyses below to optimize the GALA training routine, whereas the test split was set aside for evaluating model performance and interpretability.

## 4.2 TRADE-OFF BETWEEN RECOGNITION PERFORMANCE AND CLICKME MAP PREDICTION

We investigated the trade-off between maximizing object categorization accuracy and predicting ClickMe maps (i.e., learning visual representations that are consistent with human observers). We performed a systematic analysis over different values of the hyperparameter $\lambda$, which scaled the magnitude of the ClickMe map loss (Eq. 1), while recording object classification accuracy and the similarity between ground-truth ClickMe maps and model attention maps (Fig. 7 in Appendix). This analysis demonstrated that both object categorization and ClickMe map prediction improve when $\lambda = 6$. We used this hyperparameter value to train GALA-ResNet-50 with ClickMe maps in subsequent experiments.

## 4.3 MODEL ACCURACY

We compared model performance on the test split of the ClickMe dataset (Table 1). Here, we report classification accuracy and the fraction of human ClickMe map variability explained by feature maps at the model layer where GALA was applied (where 100 is the observed inter-participant reliability; see Appendix). High scores on explaining human ClickMe map variability indicate that a model selects similar features as humans for recognizing objects. We found that the GALA-ResNet-50 was more accurate at object classification than either the ResNet-50 or the SE-ResNet-50. We also found that all models that incorporated attention were better at predicting ClickMe maps than a baseline ResNet-50. The most notable gains in performance came when ClickMe maps were used to supervise GALA-ResNet-50 training, which improved its classification performance and fraction of explained human ClickMe map variability.

We verified the effectiveness of ClickMe maps for co-training GALA with two controls, one of which tested the importance of detailed feature selections in ClickMe maps (see Fig. 2A for examples of how these maps emphasize certain object parts over others), while the other tested whether a GALA module is necessary for a model to benefit from ClickMe supervision. For our first control, we trained a GALA-ResNet-50 on coarse bounding-box annotations of objects (see Appendix for details on how these bounding boxes were generated). The second control tested if ClickMe maps could directly improve feature learning in ResNet-50 architectures, without the aid of the GALA module (see Appendix for details on the training routine). In both cases, we found that the GALA-ResNet-50 trained with ClickMe maps outperformed the controls (Table 3 in Appendix). In other words, the details in ClickMe maps improves GALA performance, and ClickMe maps applied directly to model feature encoding did not help performance.

As an additional analysis, we tested if ClickMe maps could still improve model performance if they were only available for a subset of the training set. We trained models on the entire ILSVRC12 training set and provided ClickMe supervision on images for which it was available (test accuracy was measured on the ILSVRC12 validation set, which was not included in ClickMe). Here too the GALA-ResNet-50 with ClickMe map supervision was more accurate than all other models (Table 4 in Appendix).

## 4.4 MODEL INTERPRETABILITY

Because GALA-ResNet-50 networks were trained "from scratch" on the ClickMe dataset, we were able to visualize the features selected by each for object recognition. We did so on a set of 200 images that was not included in ClickMe training, for which we had multiple participants supply ClickMe maps. We visualized features by calculating "smoothed" gradient images (Smilkov et al., 2017), which suppresses visual noise in gradient images. Including ClickMe supervision in GALA-ResNet-50 training yielded gradient images which highlighted features that were qualitatively more local and consistent with those identified by human observers (Fig. 8 in Appendix), emphasizing object parts such as facial features in animals, and the tires and headlights of cars. By contrast, the GALA-ResNet-50 trained without ClickMe maps favored animal and car bodies along with their context.

Our ClickMe map loss formulation requires reducing the dimensionality of the GALA attention volume $A$ to a single channel. We can visualize these attention maps to see the image locations GALA modules learn to select (Fig. 8 in Appendix). Strikingly, attention in the GALA-ResNet-50 trained with ClickMe maps, virtually without exception, focuses either on a single important visual feature of the target object class, or segments the figural object from the background. This effect persists in the presence of clutter and occlusion (Fig. 8 in Appendix, fourth and last row of GALA w/ ClickMe maps). In comparison, some object features can be made out in the attention maps of a GALA-ResNet-50 trained without ClickMe maps, but localization is weak and the maps are far more difficult to interpret. The interpretability of the attention used by these GALA modules is reported in Table 1: GALA-ResNet-50 trained with ClickMe selects more similar features to human observers than the GALA-ResNet-50 trained without ClickMe (the respective fraction of explained ClickMe map variability is 88.56 vs. 64.21, $p < 0.001$ according to a randomization test on the difference in per-image scores between the models, as in Edgington, 1964).

Attention in the GALA-ResNet-50 trained with ClickMe supervision also generalizes to object images not found in ILSVRC12. Without additional training, the model's attention localized foreground object parts in Microsoft COCO 2014 (Lin et al., 2014) despite the qualitative differences between this dataset and ILSVRC12 (multiple object categories, higher resolution than ILSVRC12; see Fig. 4 for additional exemplars). We quantified the interpretability of GALA-ResNet-50 attention on a subset of the Microsoft COCO 2017 detection challenge validation set, which contained object categories also in ILSVRC12 (over 2,000 images; see Appendix for details). On these images, we measured interpretability by calculating the intersection-over-union (IOU) of an attention map for an image with its ground truth object segmentation masks from COCO (Zhou et al., 2017). By this metric, attention from the GALA-ResNet-50 trained with ClickMe supervision was significantly more interpretable than attention from the same model without ClickMe supervision (0.26 IOU vs. 0.03 IOU, $p < 0.001$ according to a randomization test detailed in Appendix).

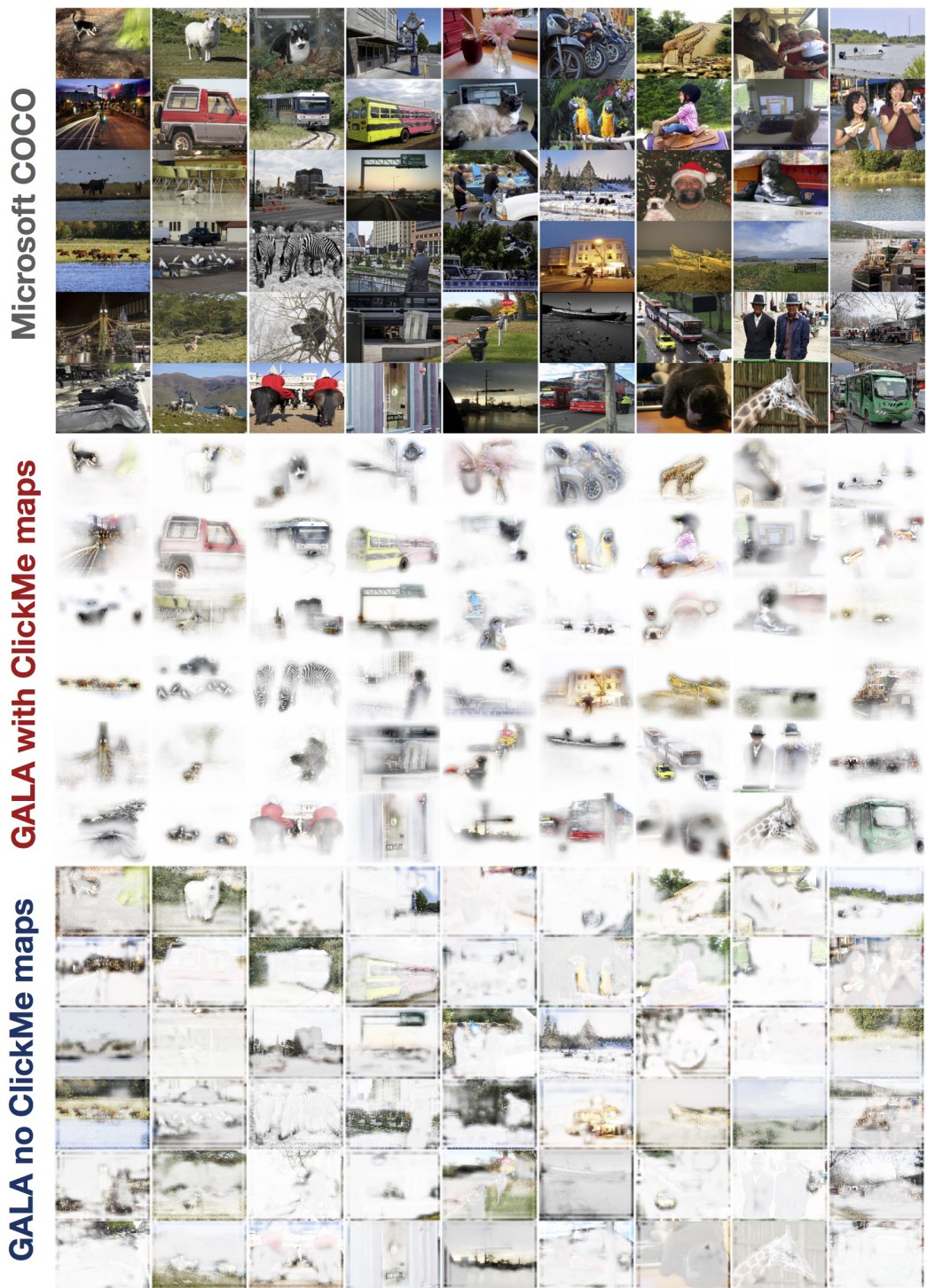

Figure 4: Exemplars from the Microsoft COCO 2017 detection challenge validation set (zoom in for detail). The top panel depicts a random subset of these images depicting object categories also present in ILSVRC12. In the middle panel, each of these images is shown with the transparency set to the attention map it yielded in a GALA-ResNet-50 trained with ClickMe map supervision. Visible features were attended to by the model, and transparent features were ignored. Animal parts like faces and tails are emphasized, whereas vehicle parts like wheels and windshields are not. The bottom panel shows the same visualization using attention maps from a GALA-ResNet-50 trained without ClickMe map supervision, which has distributed and less interpretable attention.

# 5    DISCUSSION

We have described the ClickMe dataset, which is aimed at supplementing ImageNet with nearly a half-million human-derived attention maps. The approach was validated with human psychophysics, which indicated the sufficiency of ClickMe features for rapid visual categorization. When participants viewed images that were masked to reveal commonly selected ClickMe map locations, they reached ceiling recognition accuracy when only 6% of image pixels were visible. By comparison, participants viewing images masked according to bottom-up saliency map locations did not reach ceiling performance until the full image was visible. These results indicate that `ClickMe.ai` may also provide novel insights into human vision with a measure of feature diagnosticity that goes beyond classic bottom-up saliency measures. While a detailed analysis of the ClickMe features falls outside the scope of the present study, we expect a more systematic analysis of this data, including the timecourse of feature selection (Cichy et al., 2016; Ha and Eck, 2017), will aid our understanding of the different attention mechanisms responsible for the selection of diagnostic image features.

We also extended the squeeze-and-excitation (SE) module which constituted the building block of the winning architecture in the ILSVRC17 challenge. We trained an SE-ResNet-50 on a reduced amount of data ($\sim 300K$ samples) and found that the architecture overfits compared to a standard ResNet-50. We described a novel global-and-local attention (GALA) module and found that the proposed GALA-ResNet-50, however, significantly increases accuracy in this regime and cuts down top-5 error by $\sim 25\%$ over both ResNet-50 and SE-ResNet-50. In addition, we described an approach to co-train GALA using ClickMe supervision and cue the network to attend to image regions that are diagnostic to humans for object recognition. The routine casts ClickMe map prediction as an auxiliary task that can be combined with a primary visual categorization task. We found a trade-off between learning visual representations that are more similar to those used by human observers vs. learning visual representations that are more optimal for ILSVRC. The proper trade-off resulted in a model with better classification accuracy and more interpretable visual representations (both qualitatively and according to quantitative experiments on the ClickMe dataset and Microsoft COCO images).

While recent advancements in DCNs have led to models that perform on par with human observers in basic visual recognition tasks, there is also growing evidence of qualitative differences in the visual strategies that they employ (Saleh et al., 2016; Ullman et al., 2016; Eberhardt et al., 2016; Linsley et al., 2017). It is not known whether these discrepancies arise because of differences in mechanisms for visual inference or fundamentally different training routines. However, our success in encouraging DCNs to learn more human-like representations with ClickMe map supervision suggests that improved training regimens can help close this gap. In particular, DCNs lack explicit mechanisms for perceptual grouping and figure-ground segmentation which are known to play a key role in the development of our visual system (Johnson, 2001; Ostrovsky et al., 2009) by simplifying the process of discarding background clutter. In the absence of figure-ground mechanisms, DCNs are compelled to associate foreground objects and their context as single perceptual units. This leads to DCN representations that are significantly more distributed compared to those used by humans (Linsley et al., 2017). We hope that this work will help catalyze interest in the development of novel training paradigms that leverage combinations of visual cues (depth, motion, etc) for figure-ground segregation in order to substitute for the human supervision used here for co-training GALA.

## ACKNOWLEDGEMENTS

This work was funded by ONR grant #N00014-19-1-2029. This work was also made possible by Cloud TPU hardware resources that Google made available via the TensorFlow Research Cloud (TFRC) program. We are thankful for additional support provided by the Carney Institute for Brain Science and the Initiative for Computation in Brain and Mind at Brown University. TS serves as a scientific advisor for Vium Inc, which may potentially benefit from the research results.

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

APPENDIX

ADDITIONAL CLICKME STATISTICS

The game was launched on February 1st, 2017 and closed on September 24th, 2017. Over this period, 25 contests were hosted on `ClickMe.ai` to drive traffic to the site by rewarding top-scoring players with gift cards. Participants were given usernames to track their performance and were allowed to play as many game rounds as they wanted. More than 90% of these participants played more than one image. The distribution of number of participants per image is shown in Fig. 5.

Participants' CNN partner correctly recognized object images about half of the time (47% of all images), and participants received points when this happened. In total, we recorded over 35M bubbles, producing 472,946 ClickMe maps on 196,499 unique images. Around 5% of the ILSVRC12 images used in the game were labeled by participants as having poor image quality or an incorrect class label. These are excluded from the ClickMe dataset.

ClickMe maps typically highlight local image features and emphasize certain object parts over others (Fig. 6). For instance, ClickMe maps for animal categories (top row) are nearly always oriented towards facial components even when these are not prominent (e.g., snakes). In general, we also found that ClickMe maps for inanimate objects had a front-oriented bias, with distinguishing parts such as engines, cockpits, and wheels receiving special attention.

INTER-RATER RELIABILITY OF THE CLICKME MAPS

We first verified that despite the large scale of ClickMe, the collected attention maps displayed strong regularity and consistency across participants. We calculated the rank-ordered correlation between ClickMe maps from two randomly selected players for an image. These maps were blurred with a 49x49 kernel (the square of the bubble radius in the ClickMe game) to facilitate the comparison and reduce the influence of noise associated with the game interface. Repeating this procedure for 10,000 different images and taking the average of these per-image correlations revealed a strong average inter-participant reliability of $\rho = 0.58$ ($p$ <0.001), meaning that the features that different participants bubbled during game play were very similar. We report the similarity between a model's feature attention maps and humans as a ratio of this value $\frac{\rho_{model}}{\rho_{human}}$, and refer to this as the "Fraction of human ClickMe map variability". We also derived a null inter-participant reliability by calculating the correlation of ClickMe maps between two randomly selected players on two randomly selected images. Across 10,000 randomly paired images, the average null correlation was $\rho_r = 0.18$, reinforcing the strength of the observed reliability. Below, we calculate $p$ values for correlations between model features and ClickMe maps as the proportion of per-image correlation coefficients that are less than this value.

RELIABILITY BETWEEN CLICKME AND CLICKATIONARY MAPS

ClickMe was inspired by the Clicktionary game (Linsley et al., 2017), which has two human partners play together to identify important visual features. Clicktionary experiments measured visual features for a small number of images, and each experiment included a set of 10 images to evaluate inter-experiment reliability of the selected features. We included this same set of 10 images in ClickMe in order to validate its game mechanics versus Clicktionary. The correlation between ClickMe and Clicktionary maps for these 10 images was high ($\rho_r = 0.59$, $p$ <0.001; statistical testing with randomization tests (Edgington, 1964)) and on par with the inter-experiment reliability reported for the Clicktionary game (Linsley et al., 2017). This suggests that ClickMe identifies similar visual features as Clicktionary, albeit in a much more efficient way, by swapping out one human partner with a DCN.

OBSERVERS DID NOT ADOPT A DCN-SPECIFIC STRATEGY

Importantly, participants did not adopt strategies to find visual features that were more important to their DCN partners than to other humans. In other words, participants did not carry out a "sensitivity analysis" of the features preferred by DCNs, which is impossible given the mechanics and statistics of gameplay: (1) Participants on average played fewer rounds than the number of object categories in

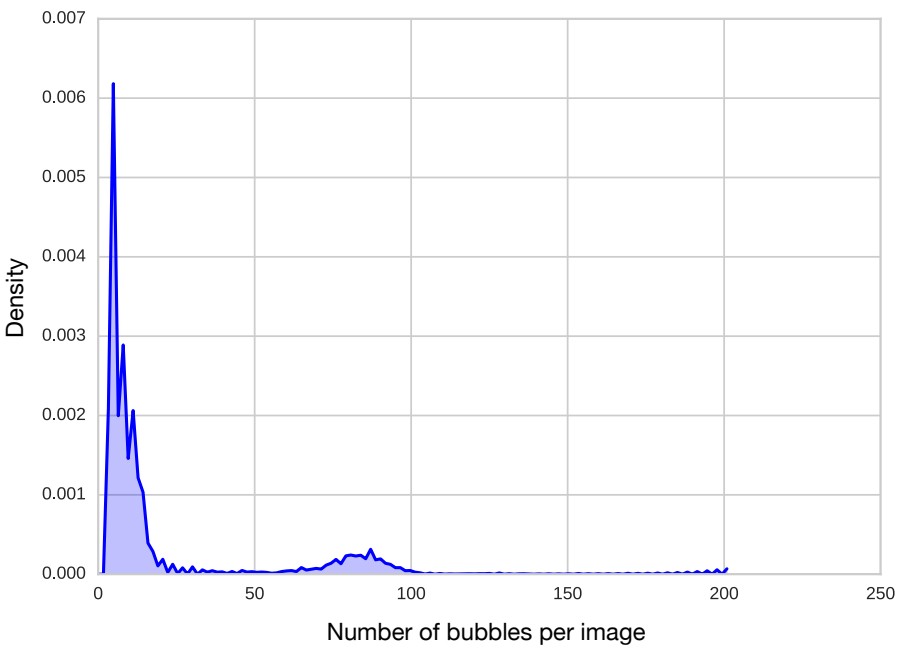

Figure 5: Distribution of participants per ClickMe map.

ClickMe (380 vs. 1,000). (2) ClickMe participants were not aware of how their clicked regions were revealed to their DCN partners (Fig. 1). (3) The top-200 most frequent players were just as likely to elicit correct responses from their DCN on the first half of their game rounds as on the second half, meaning these "expert" participants did not learn anything about features preferred by DCNs (53.64% vs. 53.61%; $t(199) = 0.04$, n.s. according to randomization test). (4) These same players did not show learning effects over a shorter timescale either: they were just as accurate on the first ten trials as they were on their second set of ten trials (49.30% vs. 52.20%, n.s. according to randomization test). In addition, as we will describe below, the similarity between ClickMe visual features selected by human participants is significantly greater than the similarity between human and DCN features.

PSYCHOPHYSICS METHODS

**Stimulus generation** We created a "stochastic" flood-fill algorithm which we applied to a phase-scrambled version of a ClickMe object image to reveal increasingly larger image regions. First, the image pixel given highest importance by a ClickMe map was identified. Second, the algorithm expanded this region anisotropically, with a bias towards pixels with higher attention scores. The revealed region was set to the center of the image to ensure that participants did not have to foveate to see important image parts and to prevent the spatial layout to affect the results. Separate image sets were generated by this procedure for ClickMe and Salicon saliency maps. Participants viewed images masked by one type of map or the other, but never both to prevent memory effects. Participants saw each unique exemplar only once in a randomly selected masking configuration. The total number of pixels in the attention maps for a given image was equalized between ClickMe and saliency maps. Original images were sampled from 4 target and 4 distractor categories: bird, zebra, elephant, and cat; table, couch, refrigerator, and umbrella.

**Psychophysics experiment** In each experiment trial, participants viewed a sequence of events overlaid onto a white background: (i) a fixation cross was first displayed for a variable time (1,100–1,600ms); (ii) followed by the test stimulus for 400ms; (iii) and an additional 150ms of response time. In total, participants were given 550ms to view the image and press a button to judge its category (feedback was provided when response times fell outside this time limit). Participants were instructed to categorize the object in the image as fast and accurately as possible by pressing the "s" or "l" key, which were randomly assigned across participants to either the target or distractor category. Similar paradigms and timing parameters yielded reliable behavioral measurements of

pre-attentive visual system processes (e.g., Eberhardt et al., 2016). The experiment began with a brief training phase to familiarize participants with the paradigm. Afterwards, participants were given feedback on their categorization accuracy at the end of each of the five experimental blocks (16 images per block).

Experiments were implemented with the psiTurk framework (Gureckis et al., 2016) and custom javascript functions. Each trial sequence was converted to an HTML5-compatible video format to provide the fastest reliable presentation time possible in a web browser. Videos were preloaded before each trial to optimize the reliability of experiment timing within the web browser. A photo-diode was used to verify stimulus timing was consistently accurate within 10ms across different operating system, web browser, and display type configurations. Images were sized at $256 \times 256$ pixels, which is equivalent to a stimulus size between approximately $5^o - 11^o$ across a likely range of possible display and seating setups participants used for the experiment.

Two participant groups completed this experiment: one which viewed images with parts revealed according to ClickMe maps, and the other with parts revealed according to their Salicon-derived salience. Statistical testing between group performance with randomization tests Edgington (1964), which compared the performance between ClickMe vs. Salicon groups at every "percentage of image revealed by feature source" bin ($x$-axis in Fig. 2). A null distribution of "no difference between groups" at that bin was constructed by randomly switching participants' group memberships (e.g., a participant who viewed ClickMe mapped images was called a Salicon viewer instead), and calculating a new difference in accuracy between the two groups. This procedure was repeated 10,000 times for every bin, and the proportion of these randomized scores that exceeded the observed difference was taken as the $p$-value.

## GALA-RESNET-50 TRAINING

In our experiments, ClickMe maps were blurred with a 49x49 kernel (the square of the bubble radius in the ClickMe game), before training to aid in convergence. Object image and ClickMe importance map pairs were passed through the network during training and augmented with random crops and left-right flips. Models were trained for 100 epochs and weights were selected that yielded the best validation accuracy. All models were implemented in Tensorflow and were trained "from scratch" with weights drawn from a scaled normal distribution. We used SGD with Nesterov momentum (Sutskever et al., 2013) and a piece-wise constant learning rate schedule that decayed by $1/10$ after 30, 60, 80, and 90 epochs of training.

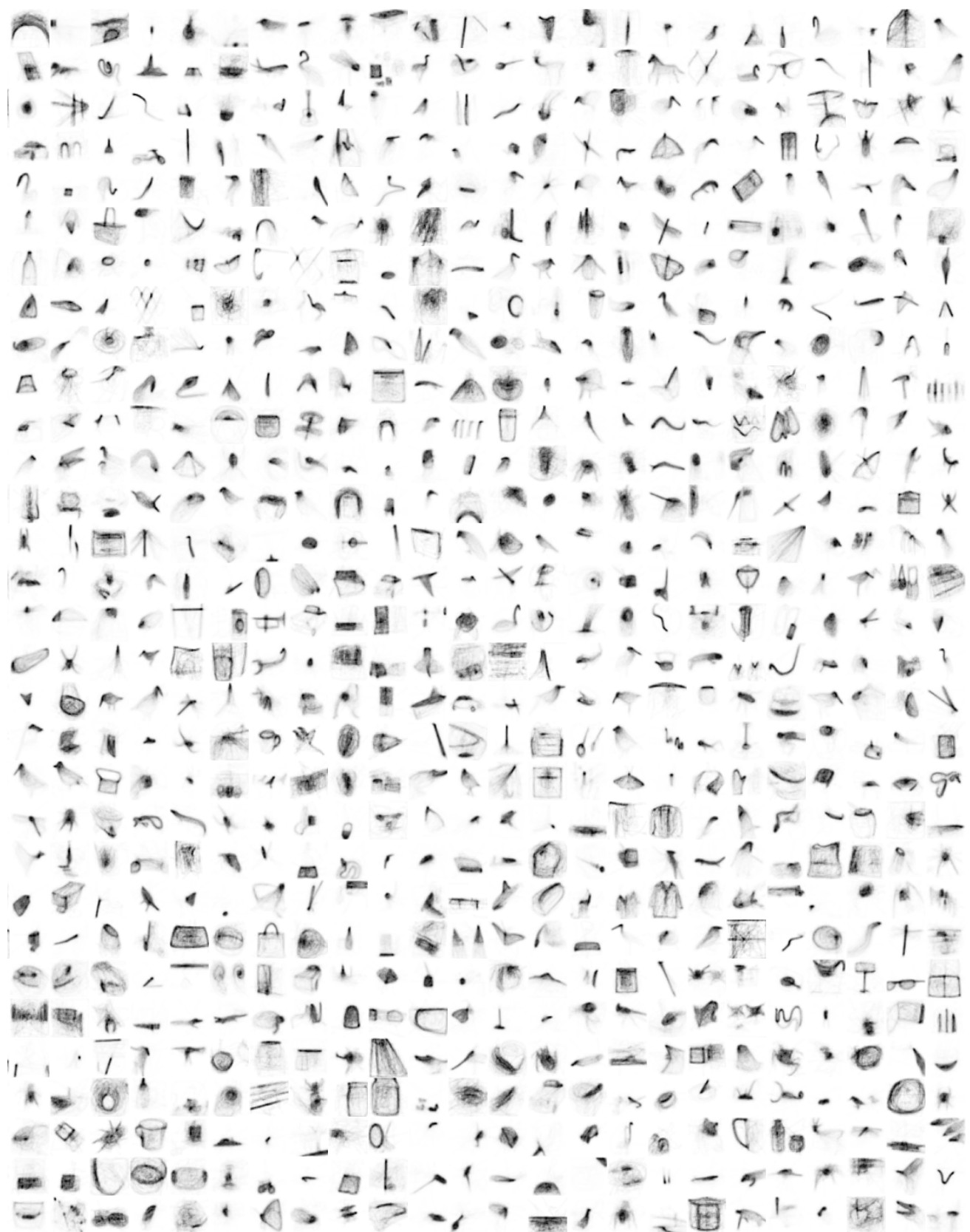

Figure 6: ClickMe map exemplars.

Models trained on full versions of ILSVRC12 (Table 2 and Table 4) were trained with Google Cloud TPU v2 devices. The large amount of TPU VRAM supported batches of 1,024 images for training. Models trained on the ClickMe subset of ILSVRC12 were trained with TITAN X Pascal GPUs (Table 1 in the main text and Table 3). Because of memory constraints, these models were trained with batches of 32 images. See `https://github.com/serre-lab/gala_tpu` for a reference implementation. Bicubic interpolation operations used for resizing ClickMe maps when training on GPUs were replaced with bilinear interpolation on the TPUs, since the former was not supported at the time of these experiments.

TRADE-OFF BETWEEN OBJECT RECOGNITION ACCURACY AND CONSISTENCY OF ATTENTION MAPS WITH HUMANS

We investigated the trade-off between maximizing object categorization accuracy and predicting ClickMe maps (i.e., learning a visual representation which is consistent with that of human observers). We performed a systematic analysis over different values of the hyperparameter $\lambda$, which scaled the magnitude of the ClickMe map loss, while recording object classification accuracy the similarity between ClickMe maps and model attention maps. Attention maps were derived from networks as the feature column-wise $L_2$ norms of activity from the final layer of GALA or SE attention (Zagoruyko and Komodakis, 2016). Model attention map similarity with ClickMe maps was measured with rank-order correlation. At each value of $\lambda$ that was tested, five models were trained for 100 epochs, and weights that optimized accuracy on the validation ClickMe dataset were selected (Fig. 7). This analysis demonstrated that both object categorization and ClickMe map prediction improve when $\lambda = 6$.[2]

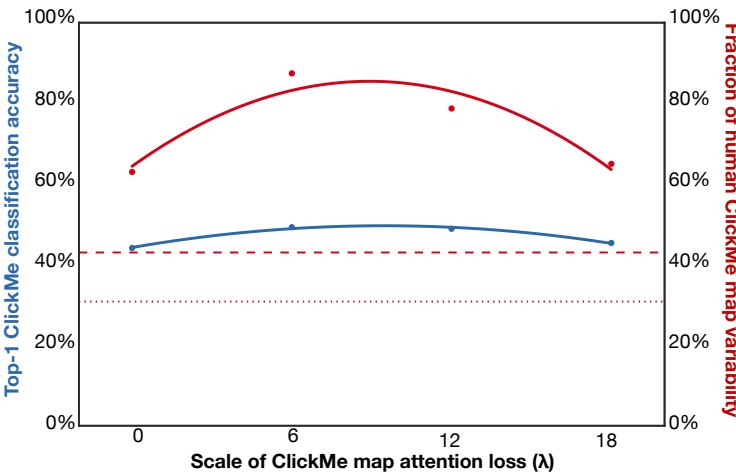

Figure 7: Training the GALA-ResNet-50 with ClickMe maps improves object classification performance and drives attention towards features selected by humans. We screened the influence of ClickMe maps on training by measuring model accuracy after training on a range of values of $\lambda$, which scales the contribution of ClickMe maps on the total model loss. A model optimized only for object recognition uses features that explain 62.96% of variability in human ClickMe maps, which is consistent with a ResNet-50 trained without attention (dashed red line). Incorporating ClickMe maps in the loss yields a large improvement in predicting ClickMe maps (87.62%) as well as classification accuracy. The fraction of explained ClickMe map variability for each model is plotted as its ratio to the human inter-rater reliability, and the dotted red line depicts the floor inter-rater reliability (shuffled null).

GALA-RESNET-50 VALIDATION

We benchmarked GALA-ResNet-50 versus a vanilla ResNet-50 and a ResNet-50 with SE attention (Hu et al., 2017) on the validation split of the ILSVRC12 challenge dataset (Table 2). These models

---

[2]We set $\lambda = 1e^{-5}$ when training on TPUs because of the different training routines for these experiments.

|  | Reference | | Ours | |
|---|---|---|---|---|
|  | top-1 err. | top-5 err. | top-1 err. | top-5 err. |
| ResNet-50 (He et al., 2016) | 24.70 | 7.80 | 23.88 | 6.86 |
| SE-ResNet-50 (Hu et al., 2017) | 23.29 | 6.62 | 23.26 | 6.55 |
| GALA-ResNet-50 no ClickMe | - | - | **22.73** | **6.35** |

Table 2: ILSVRC12 validation set accuracy for published reference models (Hu et al., 2017) and our re-implementations. Models were evaluated on $224 \times 224$ image crops from the original ILSVRC12 encoded into sharded TFRecords (`http://serre-lab.clps.brown.edu/resource/clickme`).

|  | top-1 err | top-5 err | Maps |
|---|---|---|---|
| SE-ResNet-50 | 66.17 | 42.48 | 64.36** |
| ResNet-50 | 63.68 | 40.65 | 43.61 |
| ResNet-50 w/ ClickMe | 61.32 | 41.42 | 31.18 |
| GALA-ResNet-50 w/ b. boxes | 58.14 | 35.17 | 76.42** |
| GALA-ResNet-50 no ClickMe | 53.90 | 31.04 | 64.21** |
| GALA-ResNet-50 w/ ClickMe | **49.29** | **27.73** | **88.56**** |

Table 3: Networks' classification error and fraction of explained human ClickMe map variability on $224 \times 224$ center crops from the ClickMe test set. The dataset can be downloaded from `http://serre-lab.clps.brown.edu/resource/clickme`. ** denotes $p < 0.01$ (statistical testing captures the proportion of feature maps that exceed null inter-participant reliability, detailed in *Inter-rater reliability of the ClickMe maps*).

were trained on the original version of ILSVRC12. Our implementation is consistent with published references, and we find that the GALA-ResNet-50 outperforms the other models[3].

As discussed in the main text, the GALA-ResNet-50 excelled on the ClickMe subset of ILSVRC12 (Table 3). The GALA-ResNet-50 was more accurate and better able to predict human attention maps than either the SE-ResNet-50 or the ResNet-50. This model's performance was improved dramatically when it was co-trained with ClickMe maps, which cut down top-5 error by $\sim 25\%$ versus both ResNet-50 and SE-ResNet-50.

To understand the effectiveness of ClickMe supervision when it is only available for a subset of all images in a dataset, we also tested these models on a preparation of the full ILSVRC12 for the ClickMe game. Once again, the GALA-ResNet-50 trained with ClickMe maps outperformed all other models in classification performance (Table 4).

GALA-RESNET-50 CONTROLS

Two control experiments evaluated (1) the effectiveness of ClickMe maps for attention supervision, and (2) whether attention modules were even needed for a model to benefit from ClickMe map supervision.

We first measured the importance of fine-grained annotations in ClickMe maps for supervising GALA attention. To do this, we compared the GALA-ResNet-50 with ClickMe maps to one trained on "bounding boxes" derived from these maps. Bounding boxes were created by drawing a rectangle over the ClickMe map that spanned its spatial extent. In practice, these bounding boxes do not necessarily line up neatly with classical bounding boxes drawn around objects for localization tasks. However, it still provides useful information about the resolution at which attention supervision is needed. The GALA-ResNet-50 trained with ClickMe maps outperformed one trained with these bounding boxes on both the ClickMe subset of ILSVRC12 (Table 3) and our preparation of the ILSVRC12 (Table 4).

---

[3]We observed an identical pattern of results and approximately equal performance for both the classic pre- and more recent "post-activation" versions of ResNet-50 (He et al., 2016).

|  | top-1 err | top-5 err |
|---|---|---|
| ResNet-50 | 28.60 | 9.65 |
| SE-ResNet-50 | 27.52 | 8.91 |
| GALA-ResNet-50 no ClickMe | 27.28 | 8.78 |
| GALA-ResNet-50 w/ b. boxes | 27.24 | 8.80 |
| GALA-ResNet-50 w/ ClickMe | **26.17** | **8.12** |

Table 4: Networks trained on the full ILSVRC12 training set, with ClickMe supervision on only a subset of these images (~16% of all images). This experiment demonstrates that ClickMe maps are not needed for all training samples to benefit from ClickMe supervision. Network classification error is reported on $224 \times 224$ center crops from ILSVRC12 validation[5]. The dataset can be download at http://serre-lab.clps.brown.edu/resource/clickme.

We also tested if ClickMe maps can directly supervise feature learning in residual networks. This involved replacing the ClickMe map attention loss with one that minimized the $L_2$ distance between ClickMe maps and activity volumes from a ResNet-50 during object classification training (this loss was applied to the same layers as the attention models above; Table 3, ResNet-50 w/ ClickMe). This model performed comparably to a normal ResNet-50, but was less accurate than the GALA-ResNet-50 with ClickMe maps.

INTERPRETABLE ATTENTION

We measured the interpretability of attention maps employed by GALA-ResNet-50 models trained with versus without ClickMe map supervision on a subset of images in the Microsoft COCO 2017 object detection challenge that contained animal and vehicle categories also present in ILSVRC12 (2,055 images; this criteria was used because these models were trained on ILSVRC12). For each of the selected images, we also collected animal and vehicle segmentation masks from the challenge. Thus, this amounted to a large "zero-shot" test of the interpretability of GALA model attention.

Each COCO image was resized to $480 \times 640$ pixels, and passed through the GALA-ResNet-50 trained with ClickMe and the GALA-ResNet-50 trained without ClickMe. Attention activities were extracted from the final GALA module in each module, and as was described in the main text, processed for visualization by setting attention columns at every spatial location to their $L_2$ norm, transforming the $H \times W \times C$ (corresponding to volume height, width, and channels) attention volume to $H \times W \times 1$. Because of subsampling in the ResNet-50, the height and width of the resulting attention mask was less than the input images. This was corrected by resizing these maps to $480 \times 640$ pixels.

We first visualized attention maps from the GALA-ResNet-50 models by using them as the transparency channel for their corresponding COCO images (a random subset is depicted in Fig. 4). There are striking regularities in the features selected by GALA attention when it is trained with ClickMe maps: important animal parts, such as faces and tails are consistently emphasized; as are vehicle parts like windshields and wheels (Fig. 4, middle mosaic). By contrast, GALA attention trained without ClickMe maps is much more distributed and less interpretable on these images, and highlights a combination of background and foreground elements (Fig. 4, bottom mosaic).

We quantified attention map interpretability with an approach inspired by Zhou et al. (2017), which was used to quantify the interpretability of deep network representations. For each image, we normalized attention maps from the two GALA models to the range of $[0, 1]$, then thresholded these maps to only include pixels with a score greater than 0.5. Attention maps from the GALA models with and without ClickMe map supervision were then processed to have the same number of above-threshold activities, to support a fair comparison of the two models. The interpretability of these above-threshold activities for an image was evaluated by calculating an intersection-over-union (IOU) score with the image's COCO segmentation mask, which measured the likelihood that a model's attention selected animal or vehicle parts instead of background locations. Applying this procedure to all 2,055 COCO images in our experiment yielded per-image IOU scores in the range of $[0, 1]$. Comparing the average score of the two models revealed that ClickMe map supervision significantly improved the interpretability of GALA-ResNet-50 attention: GALA-ResNet-50 trained with ClickMe map supervision interpretability is 0.26, whereas GALA-ResNet-50 trained with

ClickMe map supervision interpretability is 0.03. The difference in interpretability between the two models is statistically significant according to a randomization test (Edgington, 1964), in which the observed average difference is compared to a null distribution of average differences constructed by randomly flipping the signs of the per-image difference scores over 10,000 iterations ($p < 0.001$).

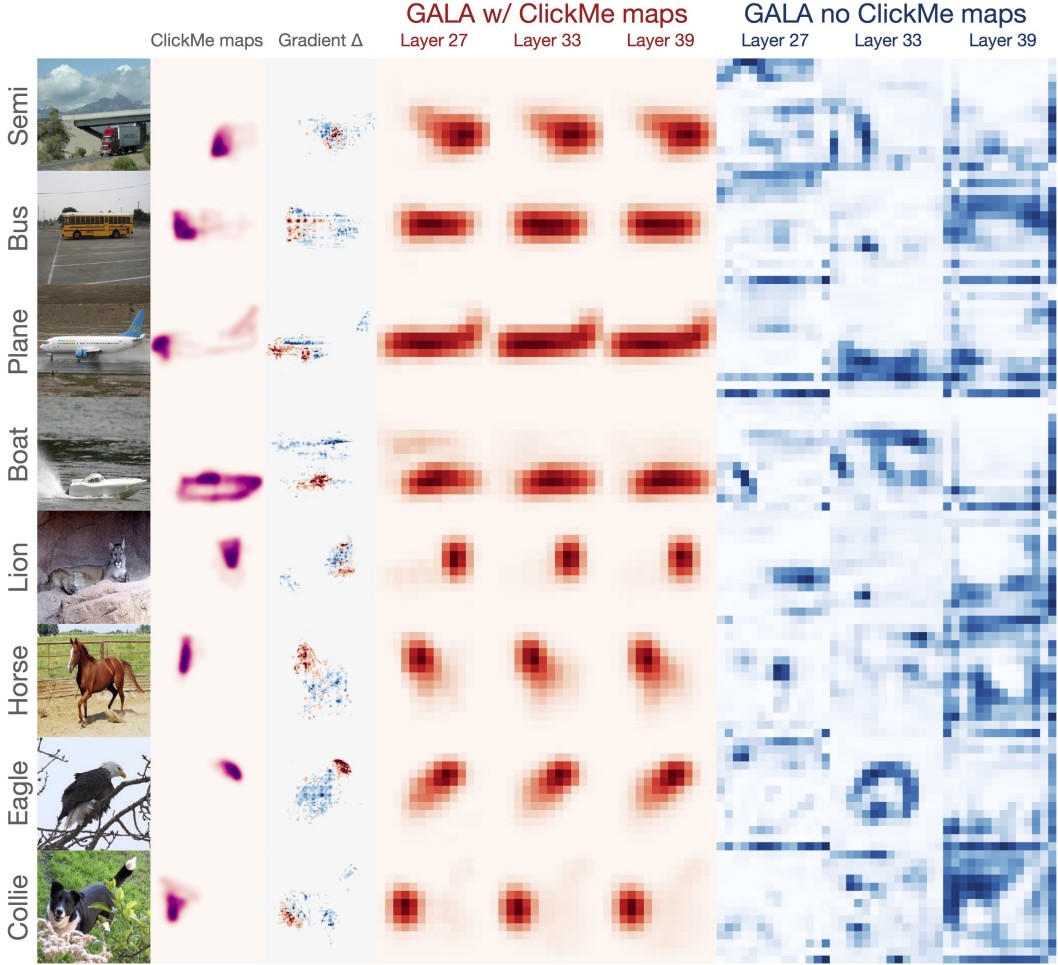

Figure 8: A GALA-ResNet-50 trained with ClickMe supervision uses visual features that are more similar to those used by human observers than a GALA-ResNet-50 trained without such supervision on images held out from model training and validation. ClickMe maps highlight object parts that are deemed important by human observers. The difference between normalized smoothed gradient images (Smilkov et al., 2017) from each network shows relative feature preferences between networks (Gradient $\Delta$). Image pixels preferred by GALA-ResNet-50 with ClickMe are red, and those preferred by a GALA-ResNet-50 without ClickMe are blue, depicting the preference for local features of the former over the latter. The column-wise $L_2$ norm of each network's GALA modules reveals highly interpretable object and part-based attention for the ClickMe GALA-ResNet-50 (in red) vs. less interpretable and more diffuse attention for the vanilla GALA-ResNet-50 (in blue).

