# OpenReview forum: "Learning what and where to attend"
_ICLR.cc/2019/Conference_

### Official Review · AnonReviewer1 · 2018-10-22
**Review of "Learning what and where to attend with humans in the loop"**

**Rating:** 8
**Confidence:** 3

**Review:**

This paper proposes a new approach to use more informative signals (than only class labels), specifically, regions humans deem important on images, to improve deep convolutional neural networks. They collected a large dataset by implementing a game on clickme.ai and showed that using this information results in both i) improved classification accuracy and ii) more interpretable features.

I think this is good work and should be accepted. The main contribution is three fold: i) a publicly available dataset that many researchers can use, ii) a network module to incorporate this human information that might be inserted into many networks to improve performance, and iii) some insights on the effect of such human supervision and the relation between features that humans deem important to those that neural nets deem important.

Some suggestions on how to improve the paper:
1. I find Sections 3 & 4 hard to track - some missing details and notation issues. Several variables are introduced without detailing the proper dimensions, e.g., the global feature attention vector g (which is shown in the figure actually). The relation between U and u_k isn't clear. Also, it will help to put a one-sentence summary of what this module does at the beginning of Section 3, like the last half-sentence in the caption of Figure 3. I was quite lost until I see that. Some more intuition is needed, on W_expand and W_shrink; maybe moving some of the "neuroscience motivation" paragraph up into the main text will help. Bold letters are used to denote many different things - in  Section 4 as a set of layers, in other places a matrix/tensor, and even an operation (F).

2. Is there any explanation on why you add the regularization term to every layer in a network? This setup seems to make it easy to explain what happens in Figure 4. One interesting observation is that after your regularization, the GALA features with ClickMe maps exhibit minimal variation across layers (those shown). But without this supervision the features are highly different. What does this mean? Is this caused entirely by the regularization? Or there's something else going on, e.g., this is evidence suggesting that with proper supervision like human attention regions, one might be able to use a much shallower network to achieve the same performance as a very deep one?

3. Using a set of 10 images to compute the correlation between ClickMe and Clicktionary maps isn't ideal - this is even less than the number of categories among the images. I'm also not entirely convinced that "game outcomes from the first and second half are roughly equal" says much about humans not using a neural net-specific strategy, since you can't rule out the case that they learned to play the game very quickly (in the first 10 of the total 380 rounds).

4. Title - this paper sound more "human feedback" to me than "humans-in-the-loop", because the loop has only 1 iteration.  Because you are collecting feedback from humans but not yet giving anything back to them. Maybe change the title?

---

> ### Author Response · Authors · 2018-11-10
> **Response**
>
> Thank you for the detailed comments and the very thorough review! Below are our responses to your suggestions on improving the paper.
>
> 1. We are overhauling Sections 3 and 4 to fix notation issues, improve readability, and clarify the figure. Along these lines and as you suggested, we will include a brief description of the GALA at the beginning of Section 3. The W_expand and W_shrink operations are borrowed from the manuscript of the original Squeeze-and-Excitation [1] module. We will revamp our description of these, which will also incorporate more of the neuroscience motivation.
>
> 2. The regularization term forces attention maps in the network to be similar to human feature importance maps. We agree that this is why the maps for different layers in Fig. 4 look similar vs. the attention maps from a GALA trained without such constraints, which are distinct. We felt that the improved interpretability, performance, and similarity to human feature maps that fell out of using this attention supervision justified its use at each layer. We also agree that the right pairing of properly supervised attention with a much shallower network could yield a far more parsimonious architecture for problems like object recognition than the very deep and very powerful ResNets.
>
> 3. We agree that the image dataset we used to compare ClickMe with Clicktionary maps is far from ideal, and we will note this in the manuscript. However, these were the only images available for such an analysis. Although it is underpowered, this analysis is also consistent with the other results we report about how the feature importance maps derived from these games are highly consistent and stereotyped between participants (section 2).
>
> Also, you raise a good point about the split-half comparison we use to demonstrate that participants do not learn CNN strategies in ClickMe. However, such a strategy would amount to a sensitivity analysis of the CNN without knowing how much of the image it was looking at: expanded versions of the bubbles placed by human players were used to unveil those regions to the CNN. The average CNN performance of 53.64% in the first half vs. 53.61% in the second half of participants' trials also does not suggest an effective sensitivity analysis. We will perform another analysis of participant performance to see if learning took place within the first tens of trials, and report this in the manuscript.
>
> 4. This is a good point. How about: “Learning what and where to attend with human feedback”
>
> [1] Hu J, Shen L, and Sun G. Squeeze-and-excitation networks. The IEEE Conference on Computer Vision and Pattern Recognition (CVPR), 2018.

---

> ### Author Response · Authors · 2018-11-25
> **Details on the new draft**
>
> In this newest draft we have overhauled explanations and readability of the entire manuscript. We have also fixed the notation issues you raised and included a clearer description of the operations of GALA. We performed another analysis of participant learning on the ClickMe game, as suggested, and found no difference in performance on the first ten versus the second set of ten trials (49.30% vs. 52.20%; this result is now included in the Appendix). Finally, we have removed the “human-in-the-loop” description of GALA training with ClickMe maps. We have also changed the title of the manuscript to: “Learning what and where to attend.”

---

### Official Review · AnonReviewer3 · 2018-11-04
**An interesting and relevant paper with poor justification for study design and analysis**

**Rating:** 6
**Confidence:** 4

**Review:**


SUMMARY

This paper argues that most recent gains in visual recognition are due to the use of visual attention mechanisms in deep convolutional networks (DCNs). According to the authors; the networks learn where to focus through a weak form of supervision based on image class labels. This paper introduces a data set that complements ImageNet with circa 500,000 human-derived attention maps, obtained through a large-scale online experiment called ClickMe. These attention maps can be used in conjunction with DCNs to add a human-in-the-loop feature that significantly improves accuracy.

REVIEW

This paper is clearly within scope of the ICLR conference and addresses a relevant and challenging problem: that of directing the learning process in visual recognition tasks to focus on interesting or useful regions. This is achieved by leveraging a human-in-the-loop approach.

The paper does a fair job in motivating the research problem and describing what has been done so far in the literature to address the problem. The proposed architecture and the data collection online experiment are also described to a sufficient extent.

In my view, the main issue with this paper is the reporting of the experiment design and the analysis of the results. Many of the design choices of the experiments are simply listed and not motivated at all. The reader has to accept the design choices without any justification. The results for accuracy are simply listed in a table and some results are indicated as “p<0.01” but the statistical analysis is never described. Interpretability is highlighted in the abstract and introduction as an important feature of the proposed approach but the evaluation of interpretability is limited to a few anecdotes from the authors’ review of the results. The paper does not present a procedure or measure for evaluating interpretability.

OTHER SUGGESTIONS FOR IMPROVEMENT

- The verb “attend” is used in many places where “focus” seems to be more appropriate.

- “we ran a rapid experiment”: what does rapid mean in this context?

- “the proposed GALA architecture is grounded in visual neuroscience” : this and many other statements are only elaborated upon in the appendix. I understand that page limit is always an issue but I think it is important to prioritise this and similar motivations and put at least a basic description in the main body

UPDATE

My most serious concerns have been addressed in the revised version.

---

> ### Author Response · Authors · 2018-11-10
> **Response**
>
> We really appreciate the comments and we are working to correct the issues you raised.
>
> We have devised an analysis that we hope can address your main critique, which involves measuring the similarity of the attention masks from GALA to object instance annotations using intersection-over-union (IOU), similar to [1]. We would like to note, however, that this is another flavor of an analysis that we present in the paper that we believe is an even more direct way of measuring interpretability: the similarity between attention masks and ClickMe maps, which describe visual features important to human observers. Please let us know if you have anything else in mind that would improve our argument of the interpretability of the attention maps from the GALA-ResNet-50 trained with ClickMe.
>
> To address your other comments, as we detailed to Reviewer 2, we will expand our description of the statistical tests used in the manuscript. We will also improve our justification for the experimental design, including a definition and more context for rapid visual recognition experiments. This experimental design has been used extensively in visual neuroscience (e.g., [2-3]), and we apologize for presenting it without appropriate context and motivation for why we chose it and the kinds of constraints that it places on participants to make visual decisions. Along these lines, we will add a discussion of the neuroscience inspiration of the GALA module to the main text. Finally, we chose the verb “attend” over one like “focus” because of its meaning in neuroscience and how the GALA module works, but will gladly re-evaluate the usage if you can point to where in the manuscript it does not make sense to you.
>
> [1] Bau D, Zhou B, Khosla A, Oliva A, and Torralba A. Network dissection: Quantifying interpretability of deep visual representations. The IEEE Conference on Computer Vision and Pattern Recognition (CVPR), 2017.
> [2] Thorpe S, Fize D, Marlot C. Speed of processing in the human visual system. Nature, 1996.
> [3] Serre T, Oliva A, Poggio T. feedforward architecture accounts for rapid categorization. Proceedings of the National Academy of Sciences, 2006.

---

> ### Author Response · Authors · 2018-11-25
> **Details on the new draft**
>
> In this newest draft we have expanded our explanations for experiments and results, detailed all statistical tests that were used, and incorporated a discussion of the computational neuroscience inspiration for GALA into the main text. We have also included a new analysis in which we quantify attention interpretability on images from Microsoft COCO, and emphasized our quantification of interpretability on ClickMe images.

---

### Official Review · AnonReviewer2 · 2018-11-04

**Rating:** 6
**Confidence:** 3

**Review:**

The paper presents a new take on attention in which a large attention dataset is collected (crowdsourced) and used to train a NN (with a new module) in a supervised manner to exploit self-reported human attention. The empirical results demonstrate the advantages of this approach.

*Pro*:
-	Well-written and relatively easily accessible paper (even for a non-expert in attention like myself)
-	Well-designed crowdsourcing experiment leading to a novel dataset (which is linked to state-of-the-art benchmark)
-	An empirical study demonstrates a clear advantage of using human (attention) supervision in a relevant comparison

*Cons*
-	Some notational confusion/uncertainty in sec 3.1 and Fig 3 (perhaps also Sec 4.1): E.g. $\mathbf{M} and {L_clickmaps} are undefined in Sec 3.1.

*Significance:* I believe this work would be of general interest to the image community at ICLR as it provides a new high-quality dataset and an attention module for grounding investigations into attention mechanisms for DNNs (and beyond).

*Further comments/questions:*
-	The transition between sec 2 and sec 3 seems abrupt; consider providing a smoother transition.
-	Figure 3: reconsider the logical flow in the figure; it took me a while to figure out what going on (especially the feedback path to U’).
-	It would be beneficial to provide some more insight into the statistical tests casually reported (i.e., where did the p values come from)
-	The dataset appears to be available online but will the code for the GALA module also be published?

---

> ### Author Response · Authors · 2018-11-10
> **Response**
>
> Thank you for the review and comments. We are working on fixing the issues that you raised, and believe that correcting them will greatly improve the quality of the manuscript.
>
> We are fixing the issues with notation, defining the variables that we neglected to in the original draft, overhauling our model figure, and improving the transitions between sections of the manuscripts. We thank you for pointing out that the statistical tests were unclear. We will incorporate the following test descriptions into the manuscript.
>
> For the behavioral experiment, this involved randomization tests, which compared the performance between ClickMe vs. Salicon groups at every “percentage of image revealed by feature source” bin. A null distribution of “no difference between groups” was constructed by randomly switching participants’ group memberships (e.g., a participant who viewed ClickMe mapped images was called a Salicon viewer instead), and calculating a new difference in accuracies between the two groups. This procedure was repeated 10,000 times, and the proportion of these randomized scores that exceeded the actual observed difference was taken as the p-value. This randomization procedure is a common tool in biological sciences [1].
>
> A similar procedure was used to derive p-values for the correlations between model features and ClickMe maps. As we mention in the manuscript in our description of calculating the null inter-participant reliability of ClickMe maps: “We also derived a null inter-participant reliability by calculating the correlation of ClickMe maps between two randomly selected players on two randomly selected images. Across 10,000 randomly paired images, the average null correlation was $\rho_r=0.18$, reinforcing the strength of the observed reliability.” The p-values of correlations between model features and ClickMe maps are the proportion of per-image correlation coefficients that are less than this value.
>
> [1] Edgington, E. Randomization tests. The Journal of Psychology: Interdisciplinary and Applied,1964.

---

> ### Author Response · Authors · 2018-11-25
> **Details on the new draft**
>
> In this newest draft we have reworked our descriptions of methods, changed our model schematic figure, and detailed all statistical tests. Thank you for these suggestions!

---

### Author Response · Authors · 2018-11-10
**General response to reviewers**

We thank the reviewers for their detailed and constructive comments. In this initial response, we want to acknowledge the raised critiques and present our plan for addressing them. Please let us know if you feel we have omitted anything. We believe that these revisions will greatly improve the manuscript.

To summarize, the revisions will address the following points:

1. We will clarify and improve the methods section by replacing our model figure, fixing notational issues, explaining our statistical testing procedures, and defining terms noted by the reviewers.
2. We will improve the flow and organization of the manuscript. This includes moving the computational neuroscience background to the related work, and expanding it.
3. We will improve our motivation for the experimental design, and take more care to walk the reader through the results as well as the effect of ClickMe-map supervision on attention.
4. We will include a link to a GitHub repository with Tensorflow code for the model.
5. We will add a new analysis to quantify how co-training a GALA-ResNet with ClickMe maps increases the interpretability of its attention maps.

---

### Author Response · Authors · 2018-11-25
**Revision**

We have uploaded a revision of the manuscript that addresses each of the points that we outlined in the meta response below. We would like to draw your attention in particular to a new analysis introduced in this draft, in which we quantified the “zero-shot” model interpretability of the GALA module trained with ClickMe maps on a large set of images from Microsoft COCO with a method inspired by [1]. As we mention in Section 4.4, GALA trained with ClickMe is significantly more interpretable by this metric than GALA trained without ClickMe (significance testing done with randomization tests, as is now described in the manuscript). We have also included Appendix Figure 8, which shows examples of the visual features favored by each model: the difference between the two models is dramatic. In total, we now have quantitative and qualitative evidence that GALA attention is more interpretable when it is co-trained with ClickMe on the ClickMe dataset (it explains a greater fraction of human ClickMe map variability) and on Microsoft COCO (more interpretable attention according to this new analysis).

We believe this version of the manuscript is greatly improved and we thank you all for your comments. We hope the manuscript now answers any remaining questions or concerns you may have.

[1] D. Bau, B. Zhou, A. Khosla, A. Oliva, and A. Torralba. Network Dissection: Quantifying Interpretability of Deep Visual Representations. Computer Vision and Pattern Recognition (CVPR), 2017.

---

> ### Author Response · Authors · 2018-11-25
> **Revision plus a diff with the submitted version.**
>
> We have uploaded two versions of the revision. (1) The most recent version is the revision. (2) The second-most recent version is a diff between the revision and our original ICLR submission. We hope this will help in evaluating our work.

---

### Meta-Review · Area_Chair1 · 2018-12-13

**Confidence:** 5
**Recommendation:** Accept (Poster)

**Metareview:**

This paper presents a large-scale annotation of human-derived attention maps for ImageNet dataset. This annotation can be used for training more accurate and more interpretable attention models (deep neural networks) for object recognition. All reviewers and AC agree that this work is clearly of interest to ICLR and that extensive empirical evaluations show clear advantages of the proposed approach  in terms of improved classification accuracy. In the initial review, R3 put this paper below the acceptance bar requesting major revision of the manuscript and addressing three important weaknesses: (1) no analysis on interpretability; (2) no details about statistical analysis; (3) design choices of the experiments are not motivated. Pleased to report that based on the author respond, the reviewer was convinced that the most crucial concerns have been addressed in the revision. R3 subsequently increased assigned score to 6. As a result, the paper is not in the borderline bucket anymore.
The specific recommendation for the authors is therefore to further revise the paper taking into account a better split of the material in the main paper and its appendix. The additional experiments conducted during rebuttal (on interpretability) would be better to include in the main text, as well as explanation regarding statistical analysis.